# Optimal Camera Placement to Generate 3D Reconstruction of a Mixed-Reality Human in Real Environments

**Juhwan Kim** [1] and **Dongsik Jo** [2,*]

1 Department of Electrical, Electronic and Computer Engineering, University of Ulsan, Ulsan 44610, Republic of Korea; juhwankim@ulsan.ac.kr
2 School of IT Convergence, University of Ulsan, Ulsan 44610, Republic of Korea
* Correspondence: dongsikjo@ulsan.ac.kr; Tel.: +82-(52)-2591647

**Abstract:** Virtual reality and augmented reality are increasingly used for immersive engagement by utilizing information from real environments. In particular, three-dimensional model data, which is the basis for creating virtual places, can be manually developed using commercial modeling toolkits, but with the advancement of sensing technology, computer vision technology can also be used to create virtual environments. Specifically, a 3D reconstruction approach can generate a single 3D model from image information obtained from various scenes in real environments using several cameras (multi-cameras). The goal is to generate a 3D model with excellent precision. However, the rules for choosing the optimal number of cameras and settings to capture information from in real environments (e.g., actual people) employing several cameras in unconventional positions are lacking. In this study, we propose an optimal camera placement strategy for acquiring high-quality 3D data using an irregular camera placement, essential for organizing image information while acquiring human data in a three-dimensional real space, using multiple irregular cameras in real environments. Our results show that installation costs can be lowered by arranging a minimum number of multi-camera cameras in an arbitrary space, and automated virtual human manufacturing with high accuracy can be conducted using optimal irregular camera location.

**Keywords:** 3D reconstruction; optimal camera placement; multi-cameras; virtual human; mixed reality





## 1. Introduction

Recently, there has been extensive research and development to create an immersive interactive environment by capturing objects or people in real environments using digital twin technologies, organizing them into 3D data and rendering them in virtual environments [1,2]. The captured data can be processed using a 3D reconstruction approach based on images captured using multiple cameras to provide a highly immersive experience by synthesizing real data (e.g., mixed-reality humans) in a mixed-reality space and expressing them as if they were real-world objects [3,4]. A large number of image-capturing equipment (e.g., the Red Green Blue (RGB) camera or Red Green Blue-Depth (RGB-D) camera set) and software calibration algorithms are required to generate target object (e.g., human in real environments) information with high accuracy [5]. However, using a large number of cameras to achieve a high level of visual accuracy in the generated 3D model can result in increased costs owing to the number of cameras [6]. Moreover, the computational time required to reconstruct the information acquired from multiple irregular camera devices, which need to be detected stable feature sets with captured images in arbitrary positions in a given space, is excessively long [7,8]. Additionally, even in scenarios where an irregular camera is utilized to acquire effective visual information, a method to reduce the matching error, which is the failure to fully acquire an object's 3D data information due to occlusion of the target object, is required [9,10].

To solve the problems associated with using multiple irregular cameras (multi-cameras) to generate 3D data for a target object, the number and placement of irregular cameras must

be optimized before calibration [8]. Recommendations for irregular camera setup configuration (e.g., the initial number of cameras and position/angle settings) are initially required to automatically generate high-precision virtual 3D objects from real objects. However, the majority of existing studies have primarily provided results on various post-processing methods (e.g., hole filing) to improve the completeness of reconstructed 3D data. Optimization studies on achieving optimal visibility using the fewest cameras during the initial deployment of multiple irregular cameras is lacking [10,11]. In recent research, to capture information from real environments and accurately apply it in virtual space, typically we have used two or more cameras [12]. Also, sensing technologies are evolving to enable reconstruction using only one camera and a deep-learning network [13]. However, we usually should use multiple cameras in order to obtain high-precision capturing of datasets.

In this study, an optimal placement method for irregular cameras is proposed for capturing three-dimensional human data in real environments using multiple irregular cameras. For example, given the need to capture information at the target location, optimizing camera placement with the minimum number of cameras allows for the obtaining of a reconstructed 3D model. The proposed method can be used to accurately reconstruct real-world objects into three-dimensional virtual data, which can then be utilized to offer an immersive sense of being with virtual objects in a virtual reality (VR) or mixed reality (MR) environment.

Figure 1 depicts a conceptual representation of the proposed optimal irregular camera placement strategy for three-dimensional object reconstruction. Space refers to the information of the real environment with diverse sizes and forms, the existence of obstacles, and the location and size of the obstacles shown in Figure 1, and Target Object refers to the object targeted in the real environment that is to be reconstructed. The object contains the size, location and a number of detailed observed objects and serves as the input for the visibility verification simulation of multiple irregular cameras.

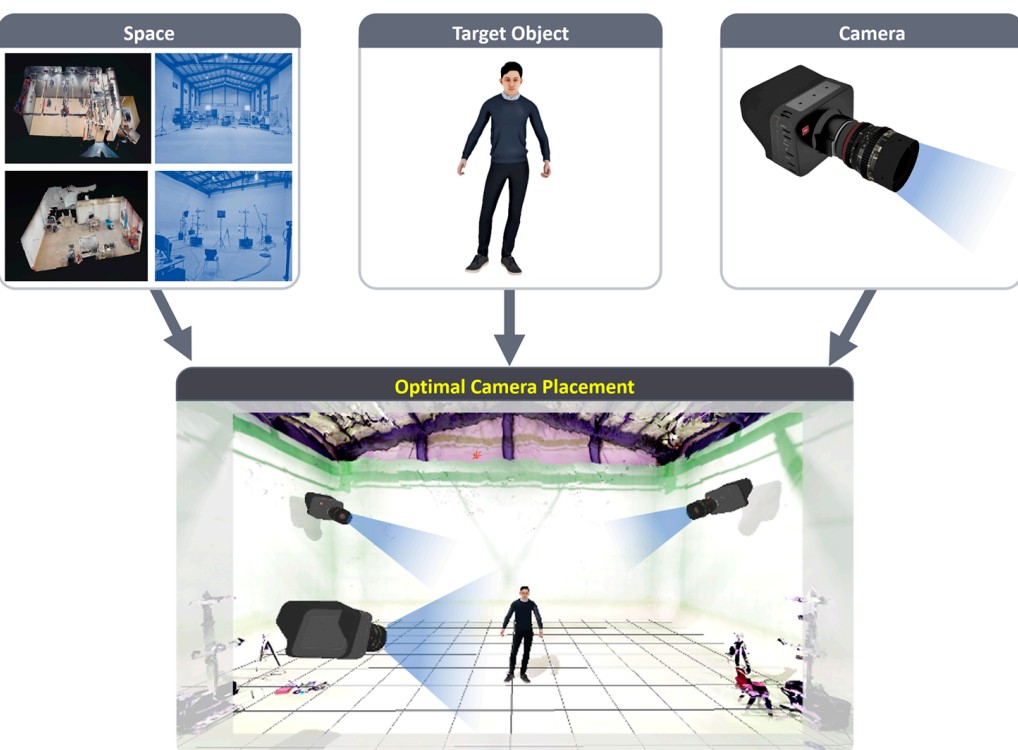

**Figure 1.** Voxel-based optimal camera placement approach. The input information (space, object, camera) is utilized to simulate the optimal camera placement to generate high-precision 3D reconstruction data of the target object in actual space.

The camera shown in Figure 1 is used to detect target objects placed in a certain space, and the placement result can be generated by considering the position, angle and number of installed cameras, and the degree of visibility provided by each camera. To further elaborate, the space is divided into voxels of varying size and spacing, and an irregular camera is placed in each voxel [14]. The angles of the irregular cameras deployed in each voxel are corrected based on the input target object to ensure that all irregular cameras have optimal visibility at their respective positions. Furthermore, the Optimal Camera Placement (OCP) stage evaluates which irregular cameras in a voxel can be deleted and removes them in succession to generate the optimal camera placement that assures visibility while employing the fewest irregular cameras in a given location. The primary objective of this study is to determine the uniformly placed cameras using voxels, gradually remove unnecessary cameras, and finally estimate the optimal camera placement that secures the visibility of the target object in a specific space to generate high-precision 3D data of the target object in the input space. Detailed information is provided in the section describing the algorithm.

The remainder of this paper is structured as follows: Section 2 addresses existing research directly related to this thesis. Section 3 provides a general description of the system, and Section 4 provides detailed descriptions. The results and analysis of the approach in this study are presented in Section 5. Finally, the overall conclusions and future work are presented in Section 6.

## 2. Related Work

### 2.1. Mixed Reality

Mixed-reality technology is a real-time computer graphics technique that combines the real and virtual worlds. Mixed-reality technology is frequently employed in a variety of fields to provide a realistic experience, providing users with a highly immersive experience in a mixed-reality area by providing fine sensory information that is similar to the experience of a particular condition in a real space [15,16]. Consequently, research is being conducted to confirm if the experience of users in a mixed-reality setting is equivalent to their experience in a real space [17]. For example, researchers developed a tele-conference system for distant cooperation by recording distant users with great precision and compositing them into the local space to create the illusion of them being in the same room [18]. The high-precision capture of real objects (or places) is required [19] to construct 3D objects from the data underlying mixed reality. In other words, mixed-reality research aims to improve the realism of virtual 3D model data provided by users via head-mounted display (HMD) devices. Thus, it is crucial to investigate how real-world object information can be preserved to accurately reconstruct modeling results [20,21].

### 2.2. Reconstruction

Reconstruction is a modern technology that constructs objects from three-dimensional digital data using information such as multiple images [22]. Generally, 3D models can be constructed by evaluating geometric structures with devices that can gather visual information, such as cameras, and several cameras are commonly used [23–25] due to the difficulty in obtaining overall structural information in three dimensions using a single camera. Three-dimensional object generation approaches that increased accuracy by securing input objects as point clouds and improving the structures are primarily investigated and developed to reconstruct three-dimensional data by utilizing numerous images (or videos) in a multi-camera scenario [26–28]. Moreover, research has been conducted to change created objects to overcome the difficulty in reconstructing a complete three-dimensional object through mapping the input information [9]. These existing research findings have primarily focused on feature point matching methods and post-processing methods based on multiple images to accurately reconstruct real objects; however, it is necessary to investigate how multiple cameras can be set up to effectively generate 3D models at the initial stage of camera installation.

*2.3. Optimal Camera Placement*

Recent research has focused on optimizing the location of many cameras, such as motion capture devices, in 3D data reconstruction [12,29]. Previous works were concerned with the quality of the 3D output depending on the installation of multiple cameras in real environments [30,31]. Currently, a method is required for carrying out camera placement utilizing a minimum number of cameras based on the visual information of the target region while minimizing the resources required for the data matching process [32–34]. This is an optimization problem for identifying objects using optimally positioned cameras, that should be simulated by considering the actual capture space, camera characteristics and objects to be captured. We propose an effective camera placement strategy for high-level 3D data capture when carrying out 3D reconstruction using multiple irregular cameras in real environments.

## 3. System Overview

Figure 2 illustrates the flowchart of the voxel-based optimal camera placement for our approach. Upon inputting the pre-reconstructed spatial information that requires optimal camera placement, the area in the space where the irregular cameras can be placed was designated and separated into arbitrarily regular areas based on voxels, with a camera designated to each voxel. Furthermore, the target object was placed in the input space in the area to be detected, and the visibility was calculated and rotated so that the target object could be best seen based on the specifications (viewing angle, etc.) of the camera. Each camera has a field of view of 40 degrees. Subsequently, the angle of the camera placed in each voxel was adjusted to have the maximum point of interest (POI) and region of interest (ROI) at the current location. The visibility of the input target object was evaluated using the detection rate of the camera, and whether the irregular camera in its current state detected the target object optimally was simulated using camera image information. Finally, in the camera estimation model stage, the camera was repeatedly eliminated from the voxels until the detected area remained just over 95%. In our case, if the detected target object was inside the camera's field of view, we considered visibility of the target information to be possible. The simulation produced the optimal camera placement in a given space for target object detection and 3D model data collecting. The experiments were conducted on a system equipped with a 13th Gen Intel(R) Core(TM) i9-13900K processor running at 3.00 GHz, 64.0 GB of RAM, and the Windows 11 operating system. The simulation environment was developed using the Unity3D engine.

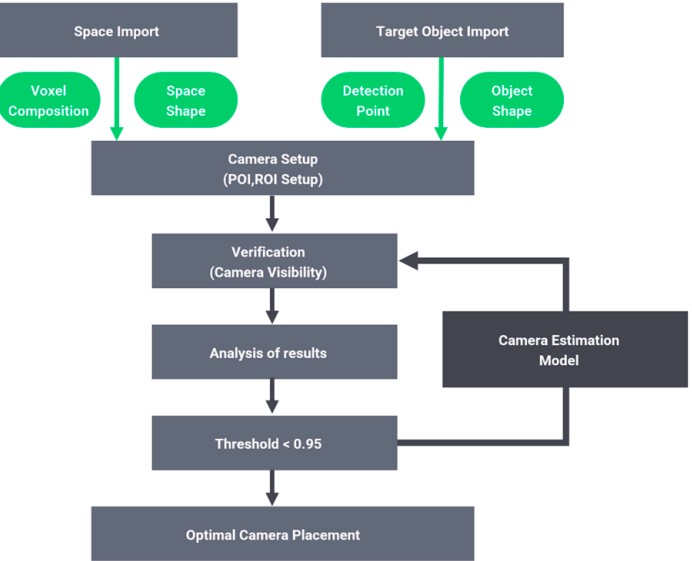

**Figure 2.** Flowchart of the proposed voxel-based optimal camera placement.

## 4. Implementation

Figure 3 shows an example of the visualization of an irregular camera placement area based on voxels. This specifies the area in which an irregular camera can be placed in the space for optimal voxel-based camera placement. The space where the camera is to be installed is evenly divided into regular areas based on voxels, and cameras are placed in each individual voxel. In other words, as shown in Figure 3, the area where an irregular camera can be placed in the input space is specified and individual areas based on voxels are also specified, enabling cameras to be placed at regular intervals. The object to be captured in the space was simulated, using the cameras installed by voxel to acquire 3D model data to estimate the optimal number and placement of cameras. At this time, the size and shape of the input space and the voxel size information for the area where cameras could be placed were defined manually.

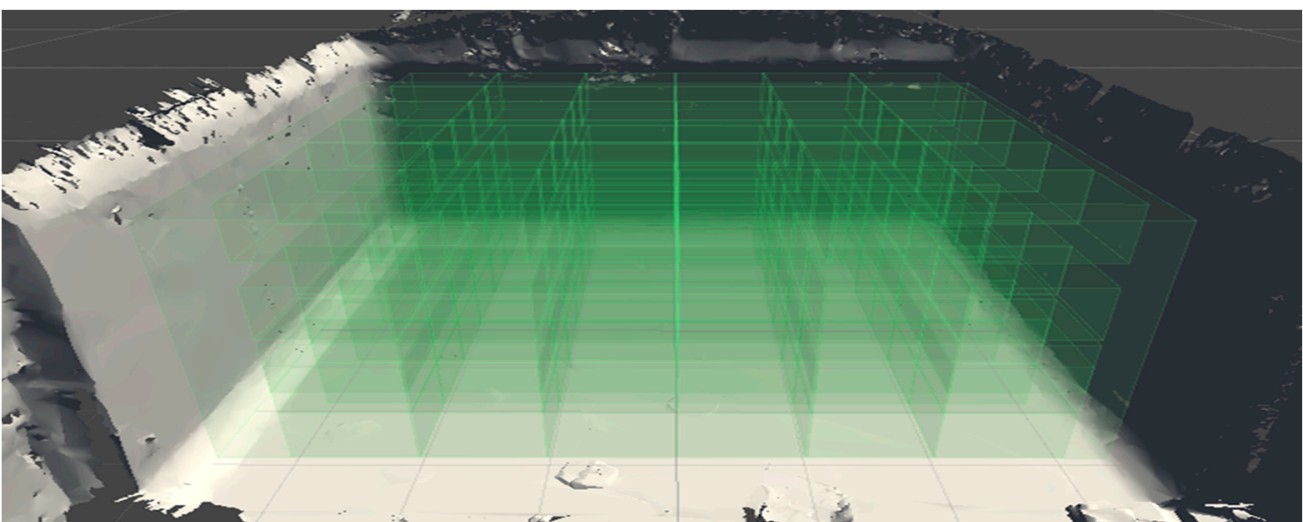

**Figure 3.** Visualizing an irregular camera placement area based on voxels.

Figure 4 depicts the outcome of pre-inputting 3D reconstruction data into the space in which the camera will be positioned. Figure 4a shows the starting state when inputting the region where the optimal camera placement is required; it includes the size of the input space as well as information regarding the probable camera placement area. Subsequently, the area where the voxels for camera placement are needed is defined, as shown in Figure 4b. Following that, the voxel where the camera is placed is defined in Figure 4c, based on the voxel region in which the user defines the camera placement point. Here, the cameras for the target object were only placed on the outskirts of the voxel area, which is where the cameras can be attached (or installed).

To derive the optimal camera placement, an item for capturing 3D model data was virtually built and placed in the area, as shown in Figure 4d, and it was set up based on the POI and ROI in relation to the location and size of the object. To summarize the implementation technique, each camera installed in each voxel region based on the placed target object is expected to ensure the greatest visibility by considering the spatial circumstances, and the camera settings are implemented to determine the spatial circumstances that maximize visibility.

Algorithm 1 shows our algorithm for the optimal camera placement that secures target visibility using the minimum number of multiple cameras in the given space. Based on POI and ROI configuration, initial camera positions and orientations are defined as a result of Figure 4d; the minimum camera setup to estimate the necessary cameras will be derived by evaluating the duplicate visibility of each camera. Camera visibility can be evaluated based on points detection for the target object.

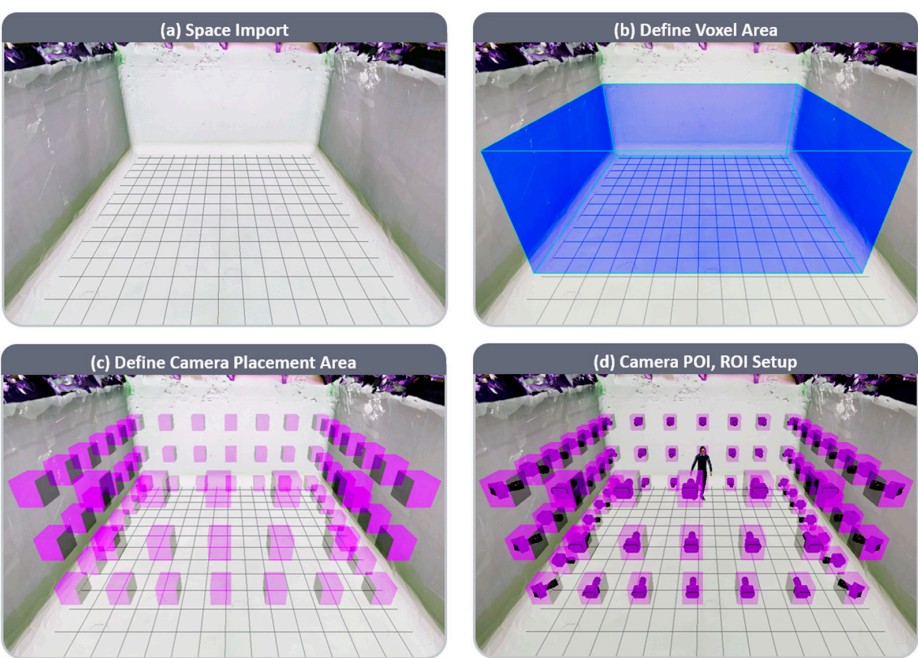

**Figure 4.** Input setup process for optimal camera placement.

Figure 5 depicts an example of input data for optimal camera placement and the simulation results for the optimal number and position of cameras to rebuild the target object. In the illustrated example, the optimal camera placement results were established such that high-quality 3D reconstruction data could be captured with a small number of cameras.

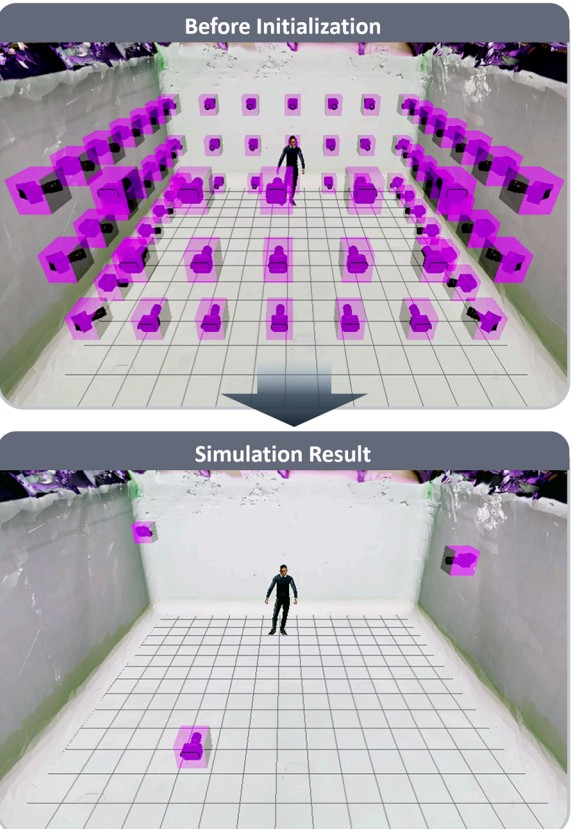

**Figure 5.** An example in terms of the results of our voxel-based optimal camera placement method.

---

**Algorithm 1:** Optimal Camera Placement Algorithm. Optimal camera placement procedure to evaluate visibility of the target object to find the minimum number of multiple cameras required.

---

**Input:** *Space, Target Object Position(T), Camera POI(P)&ROI(R) setup*
**Output:** *Optimal Camera Placement (C)*

**Function** *OCP(C, P, R, T)*
    **While** *not_empty(C)*
       *c_random = random_choice(C)*
           *// Randomly select a camera from the set of multiple cameras.*
       *V_c = {} // Visibility set for c_random*
*//Check detection point p in the field of view of camera c.*

       **For** *each point p in P*
         **If** *detect(c_random, p) <= R*
           *Add p to V_c*
       **End For**

       *V_c_prime = V_c*
       *// Define $V'(c)$ as the set of Detection Points newly detected*
       *// by $c_{random}$, excluding those already detected by other cameras.*

       **For** *each camera c_i in C − {c_random}*
       *V_c_i = {} // Visibility set for c_i*
       **For** *each point p in P*
         **If** *distance(c_i, p) <= R*
           *Add p to V_c_i*
       **End For**
       *V_c_prime = V_c_prime − V_c_i*
      **End For**
       *// Calculate $V'(c)$ by removing overlapping points from other cameras*

       *// Calculate DR_before and DR_after*
       *DR_before = union_of_all_visibility_sets(C)*
         *// Represent the Detection Rate excluding $c_{random}$ as $DR_{before}$*
       *DR_after = union_of_all_visibility_sets(C − {c_random})*
         *// Represent the Detection Rate after removing $c_{random}$ as $DR_{after}$*

       *// Check the conditions and possibly remove c_random*
         *// $DR_{after}$ is greater than or equal to the threshold T(e.g., 0.95)*
       *If DR_after >= T and is_empty(V_c_prime)*
         *Remove c_random from C*
       **End If**
    **End While**
**End Function**

---

## 5. Experiment and Discussion

We followed the steps discussed below to validate the irregular multi-camera optimization method described in this study. The input spatial information was utilized to reconstruct the real environments using a three-dimensional scan, and the area in the space where information detection was required was arbitrarily set to place the target object as a virtual person. Here, the visibility degree was measured using the camera in the voxel area to ensure the optimal visibility of the target item placed randomly in space. The method to measure visibility is based on assessing the detection points configured on the target object. The evaluation is carried out by determining how many points on the object a specific camera has detected. Our optimal camera placement aims to secure a visibility of 95% or more for a given target. Figure 6 illustrates an evaluated example of the optimal camera

placement in the input space, and we found that a high-precision reconstructed 3D model can be generated with only three camera placements.

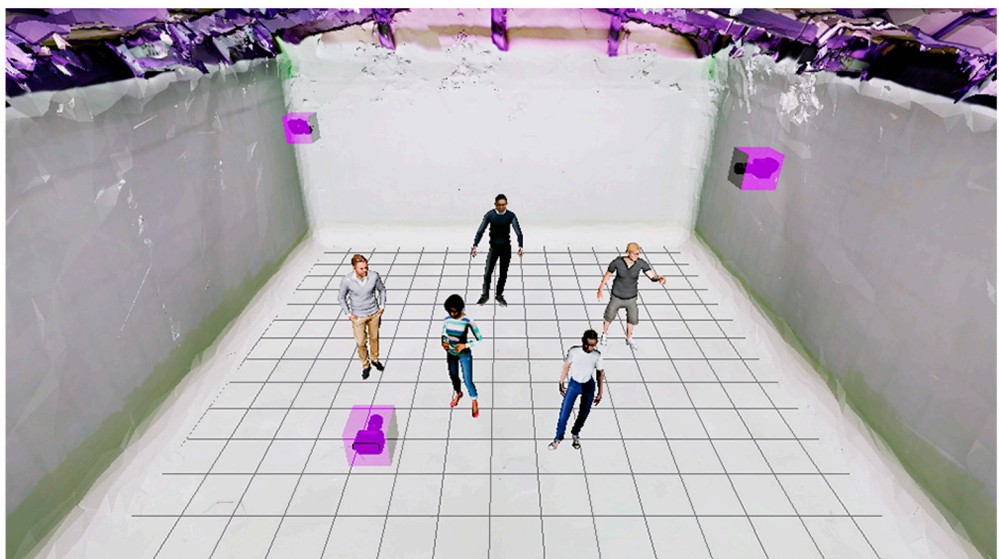

**Figure 6.** A case of visibility evaluation via various target objects' positions.

Furthermore, we conducted high-precision 3D reconstruction results to evaluate how many points were obtained in various positions of the target object. Figure 7 depicts candidate target positions to apply in our experiment. In the figure, Position_1 was designated as the initial position for evaluation, and we set up the candidates with corner positions (e.g., from Position_2 to Position_5 in Figure 7). Table 1 presents the evaluated detection rates based on the simulation results of various target inputs. To obtain high-precision 3D data in the space, the detection area should be validated by positioning target objects in various locations in the simulation stage of camera placement. As a result, the initial detection rate with the initial target object in the optimal placement for the unstructured cameras was high, on the other hand, we found that the object placed in various locations was detected at relatively lower rates. For example, for objects adjacent to the initial target object, high visibility could be secured, enabling accurate 3D reconstruction. However, it was found that objects placed at a distance relatively far from the initial target object had low performance due to visibility limitations. We will leave the establishment of methods to overcome this difficulty to future research.

**Table 1.** Evaluation results using the optimal camera placement. In the table, detection ratios of point clouds according to optimal camera placement were investigated (detection points/entire object points).

| Candidates / Camera | Camera 1 | Camera 2 | Camera 3 | Total |
|---|---|---|---|---|
| Position_1 (Initial target position) | 72/132 (55%) | 44/132 (33%) | 16/132 (12%) | 132/132 (100%) |
| Position_2 | 50/112 (45%) | 7/112 (6%) | 46/112 (41%) | 103/112 (92%) |
| Position_3 | 47/105 (45%) | 12/105 (11%) | 22/105 (21%) | 81/105 (77%) |
| Position_4 | 6/110 (5%) | 33/110 (30%) | 25/110 (23%) | 64/110 (58%) |
| Position_5 | 7/100 (7%) | 59/100 (59%) | 0/100 (0%) | 66/100 (66%) |

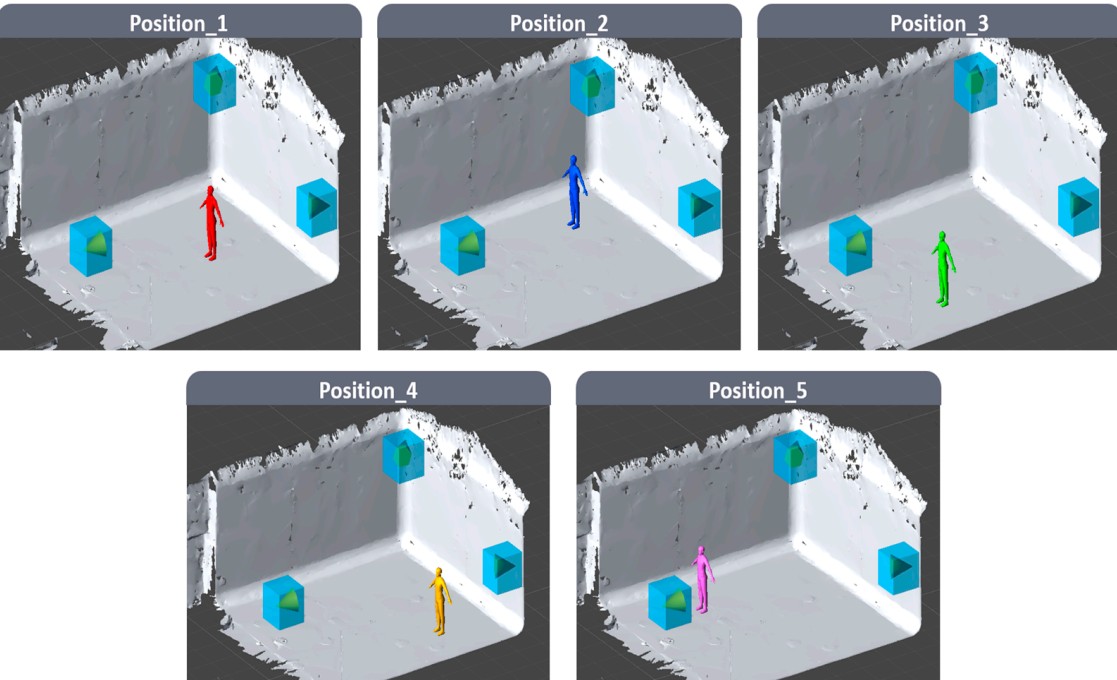

**Figure 7.** Examples of target positions that we set up as candidates: the red target at the top left was set to the initial position.

Additionally, we investigated the significance of our study with respect to the optimized method through a comparative experiment. Here, the experiment was similar to the detection ratios of point clouds (see Table 1), and we compared them to find the minimum number of cameras that would guarantee 95% visibility. Figure 8 shows the results of experimental test conditions with other methods of optimal camera placement. We conducted occlusion-based and distribution-based techniques for comparison, and measured the minimum number of cameras that could achieve over 95% visibility in reconstruction performance. Note that the occlusion-based method depends on the likelihood of dynamic occlusion for the configurations of multiple cameras in the environment, and the distribution-based method evaluated camera placement locations by optimizing the distribution of viewing angles between specific cameras and the target object [12]. In our experiment, both previous methods were manually positioned on adjacent voxels, and the angles of the cameras were also adjusted according to visibility of the target object.

Figure 8 shows our experimental results compared with other methods. Here, the number of cameras was determined with a reconstruction quality of over 95% visibility at each target location. We found that other camera placement methods required more cameras to be installed, and our approach would be helpful in reducing the number of cameras. However, Figure 9 shows examples of the limitations in our approach. Our current results need to be updated to provide more useful optimal camera placement including in many-obstacles situations in the real environment, and irregularly shaped space. We also plan to extend our approach to support everyday cases for 3D reconstruction.

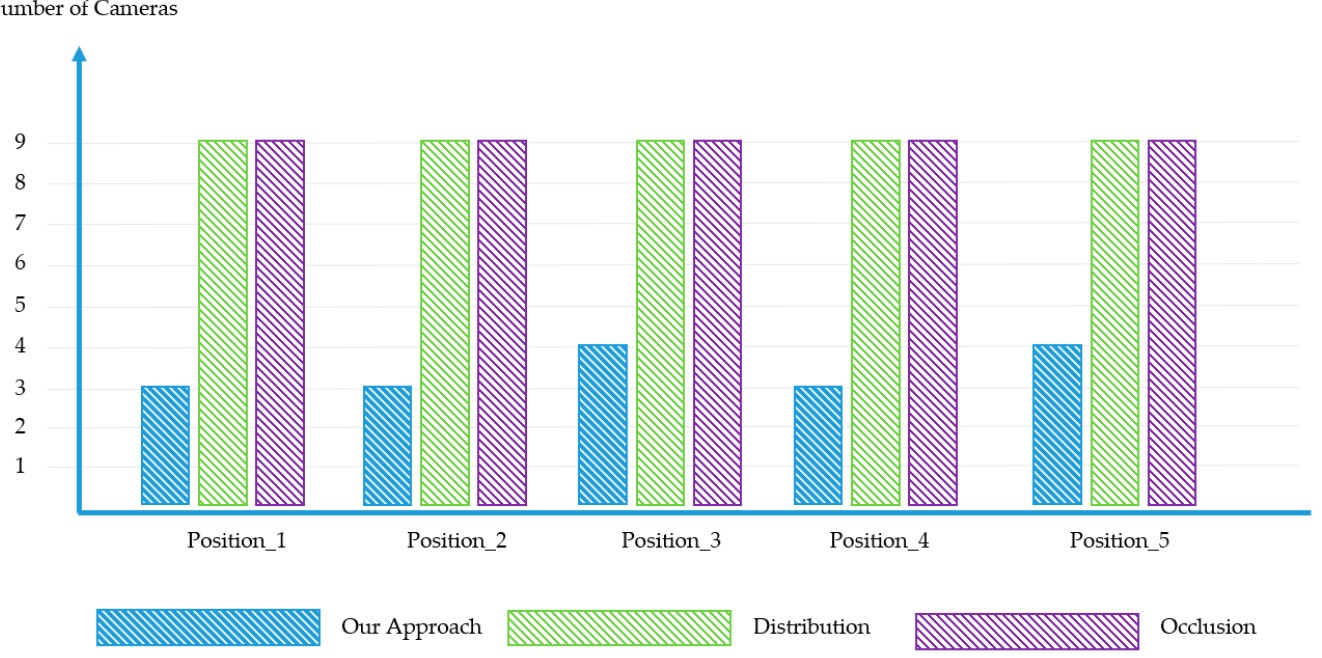

**Figure 8.** Experimental results compared with other methods: the number of cameras was determined with reconstruction quality of over 95% visibility at each target location.

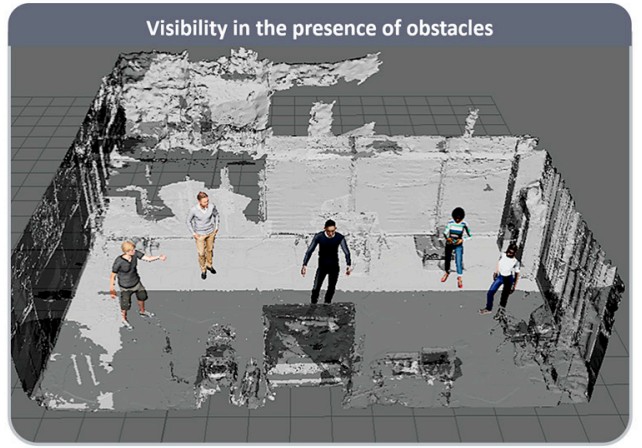 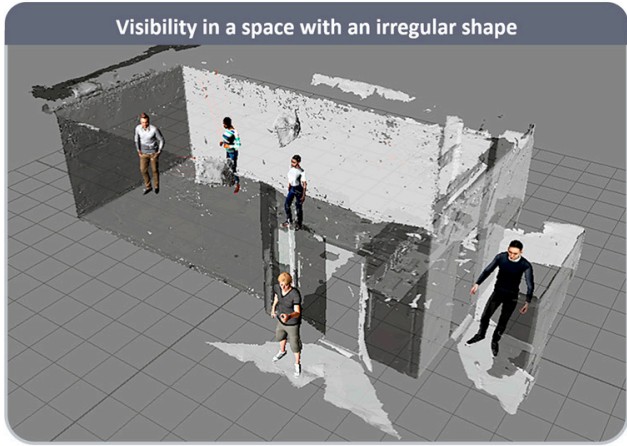

**Figure 9.** Direction for expanding the study of optimal camera placement in arbitrary space. Visibility in the presence of obstacles (**left**), and visibility in a space with an irregular shape (**right**).

## 6. Conclusions and Future Work

In this study, a method was proposed for determining the number of multiple irregular cameras required for 3D object reconstruction such as a mixed-reality human in real environments, and simulating their optimal placement to obtain high-precision data. Accordingly, we designed and tested an algorithm for optimal camera placement by partitioning the space into voxels to assure maximum visibility. We believe that our suggested method can be used to determine the optimal camera setup for high-quality reconstruction such as in remote tele-conference and motion capture systems. Furthermore, with the method proposed in this paper, we can obtain reconstructed 3D models of the real world using a minimal number of cameras based on visibility at specific points. Thus, we can optimize multiple camera configurations to minimize the camera installation costs.

However, it was assumed that the objects do not exist in a space that can generate occlusion; therefore, further research is required to address the difficulty. Furthermore, if the input space is unstructured, it may be difficult to apply the voxel placement approach

in the forward direction from where the first camera is put in, necessitating the use of an advanced algorithm. In future work, we intend to investigate the optimal camera placement strategy that can be used with the approach proposed in this study by considering the conditions that may arise in various real-world scenarios.

**Author Contributions:** J.K. implemented our optimal camera placement and conducted the experiments; D.J. organized the research and contributed to the writing of the paper. All authors have read and agreed to the published version of the manuscript.

**Funding:** This work was supported by the Institute of Information & Communications Technology Planning & Evaluation (IITP) grant funded by the Korea government (MSIT) (No. 2020-0-00226, Development of High-Definition, Unstructured Plenoptic video Acquisition Technology for Medium and Large Space).

**Data Availability Statement:** The data presented in our study are available on request from the corresponding author.

**Acknowledgments:** The authors thank the reviewers for their valuable contributions.

**Conflicts of Interest:** The authors declare no conflict of interest.

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
