# Peer review of "Optimal Camera Placement to Generate 3D Reconstruction of a Mixed-Reality Human in Real Environments"

_electronics, doi:10.3390/electronics12204244_

Round 1

Reviewer 1 Report

The motivation of this work is insufficiently described. The paper only proposes an optimization concept for camera distribution, and the specific implementation method description is not clear enough. The contribution of this work seems not clear for me. I would recommend to improve the description of motivation of the work. In addition, the author needs to add quantitative evaluation methods and results for quality improvement compared to other papers after optimization in the experimental section.

Author Response

We appreciate your thoughtful and constructive comments on our submitted paper.  We did our best to address your concerns, answer your questions and accommodate all the suggested corrections and recommendations. All the corrections or additions in the file are marked in red.

Reviewer 2 Report

The research proposes a method to optimize the camera positions in obtaining 3D human data in environments. The method involves calculating the visibility of the target in the environment space. The visibility was estimated by the detection rate of the cameras. Positions with a detection area of more than 95% was preserved. The authors conducted experiments to validate the accuracy of the proposed method. The method has sufficient novelty and it can be accepted for publication.

Author Response

(The authors gave the same response as above.)

Reviewer 3 Report

In the related works there is no comparative table of the methods or techniques used for the reconstruction of 3D data.

What are the specifications of the camera?

What is the objective or application of acquiring human data in a three-dimensional space?

The cameras manage to capture a certain work space, what dimensions do they use to acquire three-dimensional data?

They should review English, just the grammar part.

Author Response

(The authors gave the same response as above.)

Reviewer 4 Report

The authors present a optimal camera placement strategy to reconstruct 3D objects. The paper is well written and easy to follow. It can be accepted with minor revisions. 

- The authors are suggested to add experiments to compare with other OCP methods.

- Please discuss the factors that impact the proposed method. For instance, does the method still work in a complex scene with occlusions? 

Author Response

(The authors gave the same response as above.)

Round 2

Reviewer 1 Report

In comparative experiments with other methods, the number of variables should be controlled for comparison. If the results of the comparative experiments in Table 3 are all 100%, the author should demonstrate the significance of the study by comparing whether the optimized method in the paper uses fewer cameras than the 9 cameras required for the comparative experiment under 100% conditions.

Author Response

We appreciate your thoughtful and constructive comments on our submitted paper.  We did our best to address your concerns, answer your questions and accommodate all the suggested corrections and recommendations. The following is the response to the reviewer comments. we have kept the revised version of previous round in red, and updates in this round have been indicated in blue.

Round 3

Reviewer 1 Report

The author has completed the revision of the review comments